# Heart Rate Variability Measurement through a Smart Wearable Device: Another Breakthrough for Personal Health Monitoring?

**DOI:** 10.3390/ijerph20247146

**Published:** 2023-12-06

**Authors:** Ke Li, Cristiano Cardoso, Angel Moctezuma-Ramirez, Abdelmotagaly Elgalad, Emerson Perin

**Affiliations:** 1Center for Preclinical Cardiovascular Research, The Texas Heart Institute, Houston, TX 77030, USA; 2Center for Clinical Research, The Texas Heart Institute, Houston, TX 77030, USA

**Keywords:** heart rate variability, wearable device, autonomic nervous system, stress

## Abstract

Heart rate variability (HRV) is a measurement of the fluctuation of time between each heartbeat and reflects the function of the autonomic nervous system. HRV is an important indicator for both physical and mental status and for broad-scope diseases. In this review, we discuss how wearable devices can be used to monitor HRV, and we compare the HRV monitoring function among different devices. In addition, we have reviewed the recent progress in HRV tracking with wearable devices and its value in health monitoring and disease diagnosis. Although many challenges remain, we believe HRV tracking with wearable devices is a promising tool that can be used to improve personal health.

## 1. Introduction

Smart wearable devices can play an important role in daily health monitoring. As compared with traditional physician-prescribed monitors that offer only short-term recording, smart devices can provide long-term monitoring, yielding valuable datasets in different situations including exercise, sleep, or rest, regardless of age or health status. Analyzing this extensive dataset could help establish baseline heart function variables in different groups of people and evaluate variation in these variables in multiple scenarios.

In this review, we introduce heart rate variability (HRV), which measures the difference in the amount of time between each heartbeat and can be tracked by most smart devices. Because HRV is modulated by an individual’s autonomic nervous system (ANS), it can be an indicator of an individual’s health status. We describe how smart devices track HRV and the public health value of doing so. We believe HRV monitoring will provide significant insight into evaluating heart function and can further reflect ANS health status. Moreover, smart devices can be used to achieve long-term HRV monitoring by using a person’s individual information to help healthcare researchers achieve individual and precise monitoring.

## 2. Heart Rate Variability

HRV is modulated by the ANS and is a standard non-invasive marker for evaluating ANS [1], which comprises the sympathetic nervous system (SNS) and the parasympathetic nervous system (PNS). The SNS is often referred to as the “action system”, which readies the body for challenges by increasing the heart rate and blood pressure. Conversely, the PNS is responsible for the body’s recovery and relaxation after coping with challenges by decreasing the heart rate and blood pressure. Thus, the HRV value reflects the balance between the SNS and the PNS. Efficient autonomic mechanisms provide a high HRV, which indicates good adaptation to intrinsic and extrinsic factors, whereas a low HRV may indicate an abnormal, insufficient adaptation of the ANS. Therefore, measuring HRV provides an indirect reflection of the ability of the ANS to adapt to challenges that may affect an individual’s health status [2]. For example, studies have suggested that HRV is associated with stress, cardiovascular disease, diabetes, and inflammation [3,4]. Many factors such as age, sex, fitness, smoking, stress, or medication can affect HRV (Figure 1) [5]; therefore, the normal value varies significantly among individuals. The wearable device’s long-term tracking ability can provide a longitudinal HRV recording and may yield a “real” individualized normal value.

## 3. Methods

### Search Strategy

We conducted a systematic search of the relevant literature to gather information on HRV measurement through smart wearable devices. To ensure a comprehensive and thorough review, we used the following databases: PubMed, Scopus, Web of Science, and Google Scholar. The search strategy was designed to capture studies that were related to HRV and wearable technology. The primary keywords used in the search process included heart rate variability, HRV measurement, wearable devices, smartwatches, wearable sensors, and remote health monitoring. These keywords were combined using Boolean operators (AND, OR) and adjusted according to the search requirements of each database.

The inclusion criteria were as follows:Device type: studies that analyze data from commercially available wearable devices for HRV monitoring that were targeted for the personal use device market.Objective: studies with a primary objective to investigate HRV and its measurement using wearable devices.Data: studies that included reports on HRV measurements obtained from wearable devices, along with associated health or clinical outcomes.

The exclusion criteria were as follows:Outdated technology: studies that exclusively used outdated or obsolete wearable devices that do not represent current technology.Animal studies: any study conducted on animals.Objective: studies that did not focus on HRV or the use of wearable devices for HRV assessment.

An initial search in PubMed using the keywords “heart rate variability” AND “measurement” showed a total count of 26,945 articles published since 1939 (Figure 2).

## 4. HRV Metrics

Measuring HRV can be complex because it can be expressed in many formats (Table 1). Gaining a better understanding of its calculation requires breaking it down based on heart rhythm.

The first factor to consider in measuring HRV is the amount of time for recording the heartbeat, which can be recorded for more than 24 h, for 5 min (short-term), or for less than 5 min (ultra-short-term). The length of the observation can affect the results, as longer periods can capture slower fluctuations and responses to a wider range of stimuli [6]; therefore, values from different lengths of recording are not interchangeable. Varying the recording period allows for comparing HRV during multiple activities (e.g., rest, sleep, exercise).

The next step is to identify each heartbeat in the recorded data (recording methods are described later). Then, the interval between each heartbeat, known as the NN interval, can be calculated. The average of the NN intervals during the recording period provides a straightforward expression of HRV. In addition, the mean heart rate, the difference between the longest and shortest NN interval, and the difference between the nighttime and daytime heart rate can be calculated. There are also many more complex HRV expressions based on mathematical transformation of NN intervals, such as SDNN, SDANN, the SDNN index, and RMSSD (Table 1).

Having different recording times and modes of HRV expression is important because multiple internal and external factors affect HRV. For example, when considering the recording length, short-term HRV is influenced by the interaction between the SNS and the PNS and breathing pattern [7]. In contrast, the 24 h HRV is affected more by circadian rhythms, body temperature, metabolism, and sleep cycle [8]. Thus, they cannot be substituted for each other because their underlying physiological nature can be completely different [9].

Similarly, the different mathematical expressions of HRV are affected by different components of the ANS. For example, SDNN is a common indicator of both SNS and PNS activity for short-term HRV [10]. However, for 24 h HRV, the SDNN reflects more SNS activity [11]. Moreover, RMSSD and pNN50, primarily affected by the PNS, are used to indicate the ability to cope with stress [6].

**Table 1 ijerph-20-07146-t001:** Time-domain methods used to calculate HRV. Cited from [12].

Variable	Units	Description
SDNN	msec	Standard deviation of all NN intervals
SDANN	msec	Standard deviation of averages all NN intervals in all 5 min segments of the entire recording
RMSDD	msec	Square root of the mean of the sum of squares of differences between adjacent NN intervals
SDNN Index	msec	Mean of standard deviations of all NN intervals for all 5 min segments of the entire recording
SDSD	msec	Standard deviation of differences between adjacent NN intervals
NN50 count	msec	Numbers of pairs of adjacent NN intervals differing by more than 50 msec in the entire recording; three variants are possible, counting all such NN intervals in which the first or second interval is longer
pNN50	%	NN50 count divided by the total number of all NN intervals

Because the methods used to calculate HRV (Table 1) are based on the time difference, they are referred to as time-domain methods, which are easy to understand and widely used in wearable devices. Frequency-domain methods are another way to calculate HRV but are more complicated to understand because they involve the relationship between frequency and power. Frequency refers to how often a certain pattern of variation repeats over a given period, and power is the amount of variation within a frequency band. HRV frequencies can be aggregated into three leading frequency bands: a high frequency band (0.15 to 0.4 Hz, corresponding to a parasympathetic component reflecting respiration-mediated HRV), a low frequency band (0.04 to 0.15 Hz, corresponding to both sympathetic and parasympathetic components and influenced strongly by the oscillatory rhythm of the baroreceptor discharge), and a very low frequency band (0.0033 to 0.04 Hz, which may be affected by several physiological mechanisms, including the renin–angiotensin system, vasomotor tone, and thermoregulation on heartbeats) [6]. The low frequency/high frequency ratio reflects the balance between the SNS and PNS. A higher ratio may indicate more stress or alertness, whereas a lower ratio may suggest relaxation or recovery.

The power spectral density (PSD) analysis is an important method for performing the frequency-domain calculation [13]. This approach breaks down a complex signal into its basic frequency components and describes how the power (or variance) of the signal is distributed across these different frequencies. The resulting plot, called a PSD plot, typically shows frequency on the *x*-axis and power on the *y*-axis. Like the time-domain parameters, frequency-domain parameters can be calculated in different recording durations. Figure 3 is an example of a PSD analysis.

Another area of science that has contributed to the analysis of HRV is chaotic and nonlinear dynamics. The cardiovascular system is characterized by various forms of behavior, including equilibrium, periodicity, quasi-periodicity, deterministic chaos, and randomness [14], depending on its function. Fractal mathematics and chaos theory have expanded our understanding of these behaviors. Fractal geometry is evident in the physical structure of networks like blood vessels [15] and the heart’s His-Purkinje system, as well as in time-based processes like HRV variability [14].

Chaotic dynamics, characterized by positive Lyapunov exponents [16] (a mathematical concept used in the study of chaotic dynamic systems), may play a role in the complexity of HRV. In a chaotic system, the heart rate variations may exhibit a higher degree of irregularity and sensitivity to initial conditions. Lyapunov exponents may be used to analyze the sensitivity of the ANS control system, which in turn can affect HRV patterns. It is important to note that the relationship between Lyapunov exponents and HRV is a subject of ongoing research, and the application of chaos theory to HRV analysis is a complex and evolving field.

## 5. HRV Analysis Methods in Smart Devices

The initial step in analyzing the HRV is to record the heart rate and then perform the heart rate analysis based on that recording. Traditionally, these steps are performed in clinical or laboratory settings by using equipment such as an ECG, a cardiac belt, or a Holter monitor. These equipment pieces are designed for professional use and are not part of daily life practices, thus hindering the continuous and routine monitoring of HRV. With the development of smart devices, such as the Apple Watch, the Oura ring, and the Fitbit band, two methods are currently available to conveniently track heart rate in the user’s daily life: ECG and photoplethysmography (PPG).

### 5.1. ECG

The ECG function on smartwatches such as the Apple Watch and Samsung Galaxy is based on the use of one positive electrode on the back of the watch and one negative electrode at the digital crown to record a single-lead ECG. The standard procedure to record the ECG requires the user to wear the watch on the left (right) wrist and launch the ECG application on the watch by touching the digital crown with a finger on the opposite hand. This creates a bipolar signal from the voltage difference between the left and right arms, simulating the conventional ECG lead I record. Making a complete recording takes 30 s. Then, the HRV analysis could be performed based on the recorded ECG. This method can support only the ultra-short-term HRV analysis. Figure 4 shows an analysis report based on an ECG recording from an Apple Watch.

In addition to watches or bands, ECG patches or strips are available on the market. For example, the polar H10 strip with 2 ECG electrodes inside could be attached to the chest wall to continuously record an ECG for HRV analysis. Because this approach provides a much longer ECG recording period than a watch or band, it is the preferred method for long-period monitoring of HRV. ECG sensors are considered the gold standard for HRV monitoring [17].

### 5.2. PPG

PPG is a non-invasive technology that uses a light source and a photodetector at the skin’s surface to measure the volumetric variations in blood circulation [18]. Accordingly, on all smart devices, the PPG sensor is located on the back of the device so that it can contact the skin. Figure 5 shows the Apple Watch’s PPG sensor design and its working principle. When the sensor starts to work, the light source emits light to the skin tissue. The photodetector measures the reflected light from the tissue, thereby inferring changes in blood volume by measuring changes in light absorption. The light absorption is proportional to blood volume variations caused by the beating of the heart [8,18,19,20,21]. Thus, the intensity of the reflected light is different during systole and diastole. Based on this change, the pulse-to-pulse intervals determined in this manner are equivalent to the NN intervals from an ECG, and this equivalency has been verified during sleep or rest conditions in numerous studies [22,23,24,25].

The tracking quality of the PPG technique depends on good contact between the device and the skin, which can be challenging when used with watch and wristband straps, especially during periods of activity. Skin color, tattoos, and moisture can also affect PPG accuracy [27,28].

When referring to measurements of HRV that are obtained through non-invasive PPG techniques, the most appropriate terminology is pulse rate variability (PRV) [24,29], which is measured by analyzing the time intervals between consecutive pulses in the peripheral arteries, such as the wrist or finger. Although the measurement site is different from the ECG-based HRV, the underlying physiological principle is the same. The choice of terminology often depends on the measurement context and the specific tools being used for assessment.

Various factors such as stress [30], breathing patterns [31], physical activity [32], changes in posture [33], and environmental temperature [34] may affect PRV differently compared to HRV. These factors are related to changes in hemodynamics, blood pressure, or pulse transit. Because hemodynamics are primarily regulated by the ANS, it is plausible that PRV is influenced by ANS responses to external stimuli. In a study of the effect of cold temperatures on rate variability [29], hypothermia appeared to have distinct effects on PRV, not only in comparison to HRV but also when comparing different anatomical sites. Specifically, PRV tended to yield higher HRV indices, especially in cold conditions. Additionally, autonomic balance tended to be better maintained in the core vasculature. Although further investigation is warranted, the results in this study offer insight into the impact of vascular changes on PRV that are not reflected in HRV.

PPG has also been used as a tool to assess blood pressure [35], although it is typically not as accurate as traditional blood pressure measurement methods like sphygmomanometry. This measurement is accomplished via a pulse wave analysis, detecting the pulse wave of each heartbeat as it passes through a specific location, and it records the changes in blood volume with each heartbeat. The amplitude of the PPG signal can provide information about the strength of the pulse. In hypertensive individuals, the pulse may have higher amplitude due to increased pressure in the arteries. A consistently elevated amplitude may suggest hypertension [36,37].

Arrhythmia is another area of opportunity. In the context of atrial fibrillation (AF), HRV analysis has proven to be a valuable tool for several reasons. First, reduced HRV, often seen as a decrease in the standard deviation of normal-to-normal intervals (SDNN), has been correlated with an increased risk of AF development [38]. Studies have shown that individuals with diminished HRV are more prone to arrhythmia episodes, highlighting HRV’s potential as an early predictor of AF susceptibility [39]. Furthermore, HRV parameters such as the root mean square of successive differences (RMSSD) and high frequency power, reflecting parasympathetic activity, have been observed to decline in AF patients, suggesting a loss of autonomic regulation. This insight can aid in risk stratification, as patients with compromised HRV may require more intensive monitoring and intervention strategies. Finally, HRV analysis provides a non-invasive and cost-effective means to assess the efficacy of AF treatments and track disease progression over time.

## 6. Consumer Device Brands and Apps

Typically, the PPG sensor used to track the heart rate and calculate the HRV is automatically running in the background, which means the wearer does not need to perform any maneuver on the device to start or stop the recording. For example, on the Apple Watch, the HRV can be measured randomly during the day and nighttime and shows the value as SDNN in a one-minute heartbeat length. The Fitbit offers HRV monitoring during sleep, calculates the value for the whole sleep period, and presents it as the RMSDD. Table 2 shows how the major brands of wearable devices record HRV. Considering the number of smartwatches and bands sold, PPG technology has occupied much of the market.

The PPG sensors in many brands can record HRV only during sleep, whereas some can track HRV during the day and on demand (Table 2). Some research suggests that devices that use PPG techniques do not accurately track HRV during activity periods because of the susceptibility of this approach to motion noise [40] and indicates that measurements taken during rest or sleep are the most precise [41]. However, longer duration recordings that include daily activities and night or resting situations offer a more holistic measure of HRV. Morrin et al. [42] compared 24 h HRV recordings to resting 5 min measurements; they found a significantly lower coefficient of variation in 24 h measurements in all HRV parameters and superior reproducibility. Measuring the 24 h HRV is recommended for interventional studies [43]. Furthermore, when assessing the response to different stressors, measures from long-duration recordings can offer more granularity regarding the reaction of the ANS to a range of stimuli (e.g., the magnitude of the reduction in parasympathetic activity during increasing exercise intensity or before a cognitive task) [43]. Therefore, a device that can measure HRV during different situations is preferred for long-time data tracking to set up a more accurate baseline level. Currently, only ECG-based devices can provide continuous long-time HRV tracking, whereas PPG-based devices can be used for continuous monitoring only during sleep.

In each device, the expression of the analysis of HRV, such as SDNN and RMSDD, is in the apps. Most devices also provide raw data (RR intervals), which many third-party applications such as Elite HRV, HRV4training, or Welltory could access. These apps can deliver a more detailed analysis of the raw data, including most of the time-domain parameters (Table 1) and the frequency-domain parameters (Figure 6). Moreover, these apps can also be used to measure HRV in a PPG-based method, through the smartphone’s camera. This camera-based measurement is an on-demand method that requires that the user cover the phone’s camera and flashlight with a finger and keep still for several minutes to record (Figure 7). The accuracy of camera-based HRV measurement has been verified [44], giving individuals without a smartwatch or band an alternative way to track HRV.

HRV-focused biofeedback training (called heart rate variability biofeedback; HRVB) is offered through third-party apps with camera HRV tracking. Some brands (Apple Watch Breath, Oura unguided session) have on-demand measurements, and almost all ECG-based devices can be manually started and finished by users (Table 2). Briefly, HRVB is a technique that involves guided breathing at a frequency that will align the heart rate, breathing rate, and other physiological processes into a synchronous state [45]. This technique is a self-regulation therapy in which an individual can learn how to optimize ANS function. During the training session, HRV is used as an indicator to give feedback to guide the trainee in understanding how to control the body’s responses, and thus, the trainee can learn how to produce a much larger increase in HRV [46]. HRVB has been used in patients to improve many conditions, such as pain [47], anxiety [48], depression [49], hypertension [50], and even food cravings [51].

**Figure 7 ijerph-20-07146-f007:**
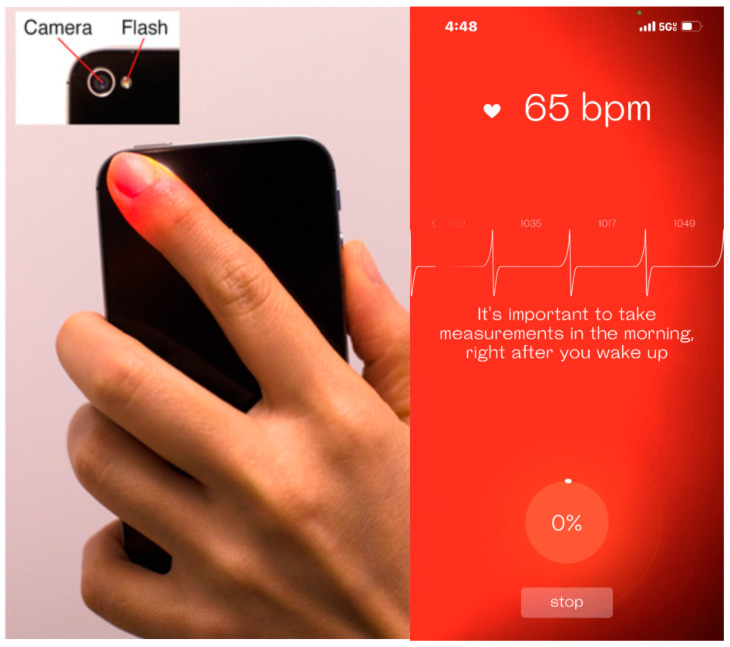
(**Left panel**) shows the use of a smartphone camera to measure the heart rate variability (HRV). Source: [52]. (**Right panel**) shows real-time phone camera based HRV recording procedures using the Welltory app on the iPhone (photo taken by the authors).

## 7. Verification of Wearable HRV Measurement

Many studies have compared HRV measurements obtained by a wearable device with those taken by a clinical ECG system [53,54,55,56,57]. A meta-analysis that included 23 studies of HRV measurements from wearable devices showed that the HRV readings had a small absolute error when compared to readings using a clinical ECG; however, this error was considered acceptable, given the practicality and cost-effectiveness of acquiring HRV through wearable devices [58]. Among the parameters used in the measurements, SDNN had the greatest amount of error, whereas RMSSD and high-frequency bands did not significantly differ in the error rates between methodologies [58]. Other verification studies have been conducted for brands such as Oura [59], Whoop [60], and Apple Watch [38], and for assessing the use of different apps to acquire camera-based HRV, including Welltory [61] and HRV4 training [62]. These studies have suggested current consumer brands and apps in the market can provide reliable HRV readings when compared with medical-level ECG measurements. However, as each piece of equipment or app has its own recording condition and new devices are released yearly, continuous verification is still critical.

## 8. Application of HRV Tracking through Wearable Devices

HRV is not a new concept and has been well studied over the past decades [59,63,64,65]. The accuracy of HRV measurements on wearable devices opens a new era for HRV tracking in the general population. In 2020, Fitbit published HRV distribution results from 8 million users based on age, time, sex, and activity; these results could be used as a framework for individual-level interpretation in future research [66]. This vast data sample can be collected primarily because of the measurement convenience and popularity of wearable devices. Collecting such a sample would be challenging using traditional ECG measurements in a clinic or hospital environment.

### 8.1. Stress

As stated previously, HRV reflects the function of the ANS as well as the balance between the SNS and PNS. The SNS releases epinephrine, which promotes rapid and widespread physiological changes such as increased heart rate. PNS generally does the opposite, such as decreasing the heart rate to promote relaxation. An overall high HRV reflects the ability of the ANS to adapt to stressors, indicating good health and optimal functional performance. Reduced HRV signifies poor ANS adaptability and is associated with fatigue, stress, and overtraining [41,43,67]. A meta-analysis of current neurobiological evidence suggests that HRV is affected by stress, and using HRV for the objective assessment of psychological health and stress may be feasible [68].

The correlation between stress and HRV makes the stress monitor function the most popular application across all wearable brands. This application increases the awareness of stress, which is important in coping with stress. However, the ANS only partially accounts for the stress response, as its role is limited only to the duration of the stressor. Another pathway, the hypothalamic–pituitary–adrenal axis [69], produces cortisol to support the SNS system and suppress other body systems, such as immune function and growth; the effects of this axis may persist for up to 90 min after the stressor ends [70]. It is possible that in some chronic stress conditions, the body may have a sustained cortisol response without specific ANS activity. In fact, some research suggests that HRV is a more direct reflection of transient physiological stress that is not aligned with perceived stress [71]. In other words, HRV is more likely to reflect the body’s instant burnout instead of stress defined as a long-term physiological response to maintain homeostasis in unexpected situations or when perceiving a threat. In a study of 657 participants wearing Garmin fitness bands, HRV was associated with perceived stress in laboratory settings; the strength of that association diminished in real-life settings. Thus, relying on wearable-derived HRV alone might not be sufficient to detect stress in natural settings and should not be considered a proxy for perceived stress but rather a component of a complex phenomenon. The study also suggested the need for longitudinal research combined with multimodality monitoring, such as sleep monitoring, to evaluate the correlation between the HRV trend and stress [72].

One advantage of wearable devices is that sleep tracking is almost a standard function and facilitates research on stress and HRV during sleep. Studies found that low HRV during sleep is associated with more mental stress [73]. Another study of the Oura ring’s HRV sleep tracking combined with a smartphone app that can log a user’s location and activity demonstrated a strong positive correlation between HRV and anxiety caused by stress [74]. The authors believe that wearable devices may provide valuable data for predicting symptoms of anxiety, most notably data related to standard measures of sleep. Continuous HRV tracking during sleep may be a promising way to further study the correlation between HRV and stress.

In another popular method for tracking and addressing stress via a wearable device, a professional trainer can help individuals cope with physical stress and improve body fitness and performance while using an activity tracker. In a comparison study of HRV acquired from a smart device and from a 12-lead EKG, smart device-derived HRVs were valid and reliable for monitoring elite athletes’ stress and recovery process [75].

### 8.2. Mental Health

Although HRV may not ultimately reflect the body’s stress, much evidence [76,77] has shown a strong relationship between HRV and post-traumatic stress disorder (PTSD). PTSD is related to fear conditioning, which results in an overactive fear response to situations that remind the individual of the original traumatic event. This triggers a robust physiological stress response involving the ANS system. In a study of 38 healthy subjects, vibration treatment (a type of low-frequency sound wave felt as a soothing vibration) from a wearable device improved HRV within 3 min under stress [78]. This vibration technique has been tested in several ongoing clinical trials [79,80,81] in which investigators aimed to improve PTSD in patients by increasing their average HRV after stress episodes.

HRV tracking can help patients improve mental health as shown by PTSD therapy. Currently, biofeedback training combined with cognitive behavior therapy [82] uses HRV as an indicator to identify the best interventional approach and as evidence of improvement after therapy to help achieve mental health resilience [83]. A meta-analysis showed HRVB improved depression symptoms in several psychophysiological conditions in adults and should be considered a valid technique to increase psychological well-being [84].

HRV tracking could also be used to monitor migraines, which are often correlated with mental health. HRV is significantly lower during the episode, suggesting the ANS is unbalanced during a migraine [85]. In a randomized experiment, HRV was significantly lower in the headache group and improved significantly after mindful practice, indicating an effective recovery after the headache [86].

### 8.3. Heart Disease

In a study in 2003 of more than 10,000 individuals [87], those with a low HRV at baseline were at increased risk of developing hypertension over 9 years of follow-up. The investigators found that a decrease in ANS function precedes the development of clinical hypertension. However, this study was based on only two HRV measurements taken 9 years apart, which does not provide a picture of continuous change between HRV and blood pressure. Some newer wearable devices are equipped with blood pressure monitoring functions, which could be a powerful tool in further examining the correlation between HRV and blood pressure and in developing better early prevention and treatment tools for hypertension. In a pilot study, smartphone camera-based HRV biofeedback training in a two-week, paced breathing intervention helped reduce heart rate and diastolic blood pressure and improve HRV in individuals with a family history of cardiovascular disease [88].

Low HRV is an established cardiovascular risk factor [89]. The association of HRV and prognosis for all-cause and cardiovascular mortality has been studied using ECG at rest, with exercise and in an ambulatory setting. In a meta-analysis, Hillebrand and colleagues used both resting and ambulatory ECG monitoring to show that a lower HRV was associated with a 32% to 45% increased risk of a first cardiovascular event in patients without known coronary artery disease (CAD) [89]. Additionally, they reported that an elevated HRV had a protective effect, with a 1% increase in standard deviation of the normalized NN interval, resulting in a 1% reduction in fatal or nonfatal cardiovascular disease events. In 2019, a large prospective international clinical study of 1043 patients showed that short-term HRV testing can be used as a novel digital-health modality for enhanced risk assessment in low- to intermediate-risk individuals without known CAD [90]; this population comprised only those with an established risk of CAD. The popularity of wearable devices will help in designing future trials for evaluating HRV and cardiovascular disease outcomes in a larger healthy population. These types of large studies will present a clearer picture of HRV’s correlation with heart health. In addition, HRV has been shown to be significantly lower in patients with heart failure [91]. Thus, HRV could be a promising tool for detecting and diagnosing heart failure, which opens a door to monitoring heart failure digitally and optimizing its management.

HRV is also a valuable index for calculating the risk stratification of sudden cardiac death (SCD) after acute myocardial infarction [92] and for predicting SCD in patients with chronic heart failure [93]. These studies suggested low HRV may be an independent predictor of increased mortality among patients with post-myocardial infarction and chronic heart failure. If HRV research could be expanded to include low-risk or general populations, investigators could assess its use as a screening tool to prevent SCD in specific groups, such as athletes or those with a significant family history of cardiovascular disease. However, this research has been limited by economic feasibility, applicability in mass screening, and comfort of the measurements. Wearable devices could solve these problems and make these studies feasible.

The larger-scale application of wearable devices will help in further investigating the correlation between HRV and heart disease in a widespread scope.

### 8.4. Cardiac Rehabilitation

Several large-scale trials and meta-analyses have documented the long-term survival benefits of cardiac rehabilitation in patients with CAD [94,95,96]. It has been recently proposed that these benefits may be due to an improvement in cardiac autonomic function; thus, HRV may be a useful gauge of cardiac function during rehabilitation [97]. Furthermore, HRV may help track the emotional experiences of patients with CAD. In a clinical trial, HRVB helped to increase ANS recovery in CAD patients who experienced negative emotions such as anger, suggesting it may be suitable for use in cardiac rehabilitation [98]. In another trial, HRVB increased HRV and decreased expressive and suppressive hostility behavior in patients with CAD after training [99]. These studies suggest that HRVB could be used as adjunct training during cardiac rehabilitation to help CAD patients manage their stress [100].

### 8.5. Diabetes

Diabetes, a metabolic disorder characterized by chronic hyperglycemia, is associated with ANS dysfunction, which correlates with decreased HRV indices. Moreover, decreased HRV in patients with diabetes is a predictor of cardiovascular morbidity and mortality. In a study of CAD patients with and without diabetes, patients who had CAD with diabetes did not show improvement in HRV after 8 weeks of cardiac rehabilitation training [101].

HRV can also be used to monitor hypoglycemia. Because hypoglycemia stimulates the SNS, HRV could be helpful for real-time early detection of hypoglycemia. In a pilot study of 23 patients with type 1 diabetes [102], HRV was continuously measured through a wearable device for 5 days. The results showed HRV patterns identified all but 18% of hypoglycemia events, which suggests that measuring real-time HRV may be useful in early hypoglycemia detection.

### 8.6. Inflammation

Since there is considerable interplay between systemic inflammation and ANS activity, HRV may be a non-invasive and easy-to-use tool for early detection of a developing systemic inflammatory response. Previous work has demonstrated that HRV analysis can identify patients developing pancreatitis [103]. In this study, HRV measurement on admission was a good predictor of necrosis and organ failure in patients with severe acute pancreatitis. In a study [104] in which HRV was recorded by a wearable device, volunteers were intravenously given lipopolysaccharide to induce systemic inflammation. Frequency-domain HRV parameters showed a significant change shortly after plasma cytokine levels increased, and the change preceded the onset of flu-like symptoms and alterations in vital signs. These findings suggest monitoring HRV may be a promising tool for the early detection of a systemic inflammation response. During the pandemic, tracking HRV via a wearable device was used to identify a COVID response. In a study of 297 participants [105], significant changes in HRV were seen before a positive PCR test from a nasal swab sample, suggesting that longitudinally collected HRV metrics from a commonly worn commercial wearable device may be able to predict the diagnosis of COVID-19 and identify COVID-19-related symptoms. In a larger study of 2745 subjects [106], similar results from data collected from wearable devices showed that HRV monitoring may improve early detection of COVID-19. A study [107] of 141 critically ill patients with COVID-19 and 209 patients with all-cause sepsis found that changes in HRV were statistically different between the two groups, suggesting HRV levels could potentially differentiate between severe COVID-19 infection and bacterial sepsis. This finding was attributed to the cardiac involvement in COVID-19 causing significant changes in HRV pattern.

## 9. Advantages of Using Wearable Devices for HRV Tracking

HRV is a valuable indicator that reflects the overall health status of patients, as described above. Previously, the Holter ECG was the principal way to perform longitudinal heart rate monitoring with HRV analysis. However, the inconvenience of wearing a Holter monitor limits its application in many situations, such as swimming. In addition, skin irritation makes long-term wearing of the device problematic. In contrast, wearable smart devices have been incorporated into an individual’s daily life and are designed to monitor health status during many different activities, including resting, sleeping, walking, running, or even swimming. By using these devices, individuals can track their health status for years. Because of their popularity and widespread use, these devices can be used for large-scale studies. In 2022, an estimated 67 million people were projected to use a wearable device in the US; 50% of consumers were interested in tracking their cardiac health, and 68% of physicians intended to use a wearable device for patient monitoring [27]. This extensive population will provide a significant sample pool that will create the opportunity to examine and solve issues previously deemed challenging to study. In addition, smart devices are closely connected to AI algorithms; therefore, monitoring and analysis can be quickly scheduled and performed, dramatically improving the accuracy of the diagnosis and user compliance. Successful examples include the multiple devices used for detecting AF, which have ushered in a new era of personal health [108,109,110]. AI can also detect new patterns that are difficult to recognize using human-based methodologies. For example, AI has recently been used to analyze single-lead ECG data from wearable devices and has predicted acute left ventricular dysfunction with an area under the curve of 0.88. This result is slightly better than a medical treadmill diagnostic test [111]. More advantages of using wearable devices to track HRV are described below.

Awareness of individual health and increased user compliance. People with and without wearable devices have shown significant differences in activity level and physical health awareness [112]. As wearable devices remain popular in the market and more apps are being developed, healthcare workers have an excellent opportunity to promote health education more efficiently. In turn, people will be more likely to follow instructions to record health data in clinical trials, which will significantly improve participants’ compliance and facilitate the performance of the trials.

Large-scale monitoring for personalized medicine. Longitudinal and large-scale monitoring to set up baseline data for individuals and populations is a pathway that will lead to the development of personalized medicine. As daily mobile electronics, wearable devices can substitute for expensive heart monitors that previously were accessible only in hospitals. Moreover, some wearable devices can be purchased for $100. The ability to automatically record data eliminates the need for trial participants to travel to research facilities and thus decreases both the cost and complexity of clinical trials. These advantages make long-term and large-scale trials possible. For example, in a recent study [113], investigators analyzed nine million HRV readings through the HRV4training app from 28,175 users over 5 years. The study monitored HRV when the user responded to different stressors, such as training, high alcohol intake, menstrual cycle, and sickness. The authors concluded that measuring HRV upon waking by using a smartphone app could effectively be incorporated into normal daily life to quantify individual stress responses across many scenarios. These types of datasets can be used to monitor HRV trends over years and establish reference points in different populations based on age, sex, race, and occupation.

In addition, wearable devices can record HRV during periods of physical activity in many different sports and in other conditions such as rest and sleep. This feature can be used to identify factors that may affect HRV and to correlate HRV changes with the user’s physical condition and activities. Moreover, these data can help direct users in improving their HRV. A higher and more consistent HRV from day to day is typically associated with better health and improved fitness [114,115]. Thus, wearable devices can serve not only as a tracking tool but also as a personal training tool to help users get positive feedback, such as in HRVB, to improve their condition. In a meta-analysis of HRV-guided training studies [116], the authors concluded that HRV-guided training may be more effective than predefined training for maintaining and improving vagal-mediated HRV, with a slight margin of improvement in fitness and performance. Similarly, another meta-analysis showed that in comparison to predefined training, HRV-guided endurance training had a medium-sized effect on submaximal physiological parameters [117].

Efficient high-complexity analyses with AI algorithms and wearable devices. A deep learning AI algorithm based on smart band HRV data (collected over 2 to 5 min periods) has shown improved mental and general health predictive capability [118]. A simulation algorithm using ensemble empirical mode decomposition-based entropy features has been studied for analyzing HRV data to predict SCD; this approach predicted subjects at risk of SCD up to 14 min before SCD onset with an accuracy of 96.1%, a sensitivity of 97.5%, and a specificity of 94.4%. Once this novel algorithm can be integrated into wearable devices, it could help in creating a protocol embedded in a mobile app to identify people with SCD risk [119]. With the help of AI, investigators can not only collect and analyze data automatically and more accurately, but also recruit patients worldwide without their needing access to the hospital. For example, Apple’s ResearchKit app helps researchers access raw data and develop clinical trials based on the iPhone, a convenient way to enroll subjects worldwide. In 2019, Apple published its first large-scale research study [110]: The Apple Heart Study collected data from over 400,000 Apple Watch users in 8 months. This is the first-ever, large-scale virtual study conducted in a real-world setting based on a phone app to recruit participants without hospital access. Based on this successful experience, Apple has developed a research app to design clinical trials. Every user can download it and easily enroll in studies they are interested in. Currently, three longitudinal trials are running in these apps: Apple Women’s Health Study, Apple Heart, and the Movement and Hearing Study. This new type of trial will facilitate data collection, contribute to large data pools across a large population, and build a personal reference for future comparison of individual trends.

## 10. Challenges

The biggest challenges to using wearable devices for tracking HRV are the poor standardization and lack of consistency in methodology among the brands. The PPG-based tracking method, which dominates the consumer market, is unsuitable for continuous monitoring because it is affected by many factors, especially the user’s activity. A few brands achieve continuous recording during sleep. Although HRV tracking in sleep is important, it cannot replace the 24 h recording for identifying long-term trends. The 24 h continuous recording could represent the cardiovascular system’s response to various environmental stimuli and workloads. Circadian rhythms, core body temperature, metabolism, sleep cycle, and the renin–angiotensin system contribute to variability in 24 h HRV recordings [7]. Hence, 24 h HRV monitoring encompasses daily and nocturnal activities, and this type of longitudinal recording may provide a more holistic measure of HRV and yield the gold standard value for HRV. Currently, only ECG-based sensors provide good continuous tracking; however, the design of these sensors, such as a strip or patch, does not allow for easy or popular daily wear. Thus, until the auto-continuous tracking technology is improved for wearable device-derived HRV measurements, it will be difficult for researchers to record reliable long-term trends in HRV, which are important for establishing a personalized baseline and making correlations between HRV and chronic conditions.

Different brands offer different solutions for solving this issue. The Apple Watch has a unique algorithm that provides intermittent automatic HRV measurements day and night, even when the user is physically active. However, the readings last only 1 min every time, which is not considered continuous tracking. In addition, the 1 min readings fall into the ultra-short-term recording category and are easily influenced by external factors during tracking, such as stress, physical exercise, coffee or alcohol intake, and hydration conditions. Without a clear log, discerning whether a reading is baseline data or reflects some external factor is challenging. Unfortunately, with this algorithm, it is unclear why the Apple Watch measures the HRV at these specific time points, making the recordings seem random. Thus, correlating the HRV with the user’s activity is difficult (Figure 8).

In addition, the data sample strategies among the brands are different. Fitbit and Oura have different HRV sample methods during sleep. In a study comparing readings from six wearable devices with ECG-based HRV, the HRV agreement varied among the brands [60] (Figure 9). If investigators do not consider this difference, they may ignore the data variations. For example, baseline data from research with the Fitbit may not show the same trend as studies using the Apple Watch. Therefore, more research is needed to compare brands to eliminate data disagreement.

These limitations are seen in many studies, as true baseline HRV measurements are challenging to collect. In most studies, the data collection is based on “snapshot” measurements (short or ultra-short records, such as seen with the Apple Watch), which could be significantly affected by external and internal factors, leading to inaccurate baseline measurements. In some studies, measurement of “baseline” HRV has been conducted close to the stimulus; therefore, responses to the stimulus may have already started to occur and would affect the HRV [120]. Because HRV is unique to each person, the accuracy of baseline HRV values is fundamental to ensuring confidence in subsequent measurements. Dobbs et al. [58] concluded that meaningful interpretations of longitudinal HRV data are improved by using weekly averages of consecutive day-to-day recordings, which are superior to snapshot measures of HRV [121]. Unfortunately, there is currently no longitudinal study using wearable devices to record HRV continuously. Thus, the issue of standardized baseline data remains and needs to be a significant consideration when designing studies or evaluating data from wearable devices.

Another hurdle is that many factors are connected to HRV. To get the best attribution, the user needs to make an accurate daily log; this task may decrease users’ compliance and prevent them from generating meaningful records. As previously discussed, the Apple Watch automatically measures HRV during the day; however, users may not be able to recall the circumstances under which the measurement was taken. To solve this issue, some brands focus on measuring HRV only during sleep; although this approach excludes many external stimuli, it cannot reflect changes in HRV during the day and thus does not provide a holistic view of HRV. Some apps use on-demand measurements such as HRVB, which asks users to take HRV measurements manually and records the user’s daily condition and activities. In this way, users can log their activity and other factors more precisely, but the approach relies heavily on the users’ cooperation (Figure 10). Therefore, it is difficult to say that HRV tracking is currently an automatic function. In the future, devices or apps must be developed to provide easier ways of helping users track their daily activities and health conditions. Oura’s activity detection, which can automatically recognize the wearer’s exercise activity, is a good step toward improving the automatic tracking function of wearable devices.

Designing large-scale longitudinal trials to achieve accurate results is challenging using current technology. Data collection should focus on short periods (1–2 weeks) of continuous tracking, using an ECG-based device in specific populations (e.g., athletes, patients who need remote monitoring). Technology is progressing daily, and we believe more mature tracking methods will soon be available.

## 11. Conclusions

HRV is a reliable indicator of ANS function. Its measurement has been built into almost every wearable device on the market, and HRV can be conveniently recorded and analyzed. Designing clinical trials to track HRV, monitor individual health, and improve mental and physical health status through HRVB is feasible. Although there are technical limitations, physicians should be aware of the importance of HRV and educate patients on its usefulness, especially considering the popularity of wearable devices. With some progress in wearable devices, large-scale and longitudinal tracking of HRV will provide valuable and precise insight into HRV trends and how they relate to health management.

## Figures and Tables

**Figure 1 ijerph-20-07146-f001:**
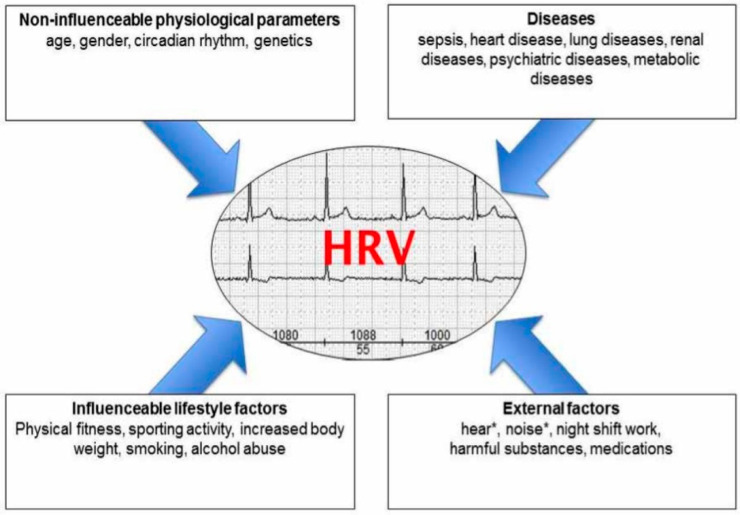
Factors that affect heart rate variability (HRV). Source: [5]. * HRV decrease as a result of a physiological reaction to a physical stimulus. Provides a summary of the results referring to the factors and covers the four main areas, i.e., non-influenceable physiological factors, illnesses, influenceable lifestyle factors, and external factors.

**Figure 2 ijerph-20-07146-f002:**
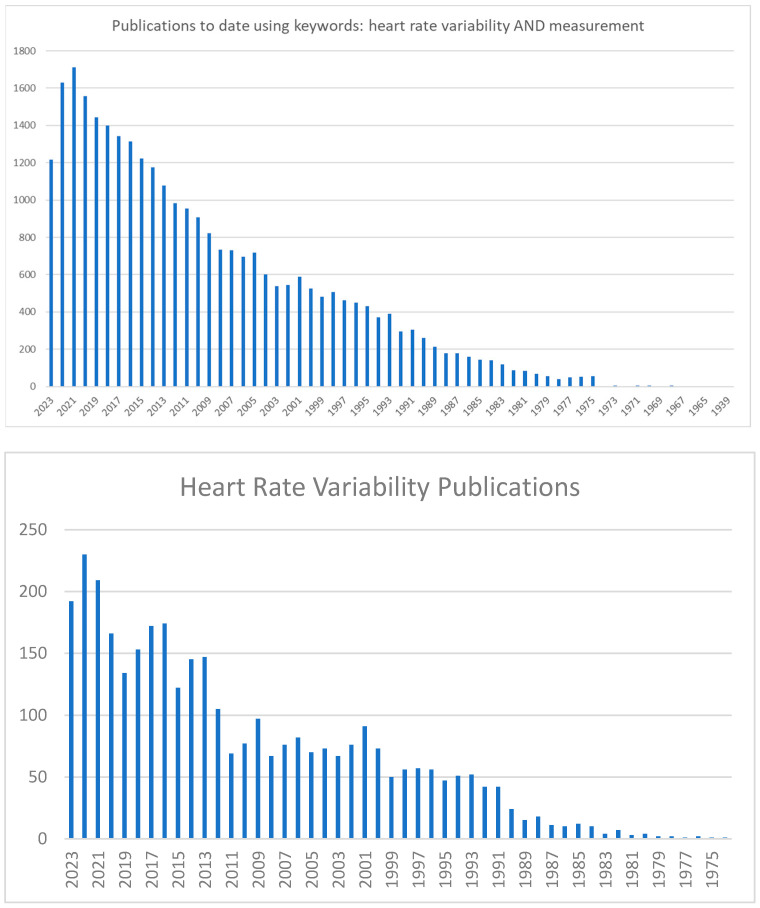
PubMed publication records when using keywords: heart rate variability AND measurement.

**Figure 3 ijerph-20-07146-f003:**
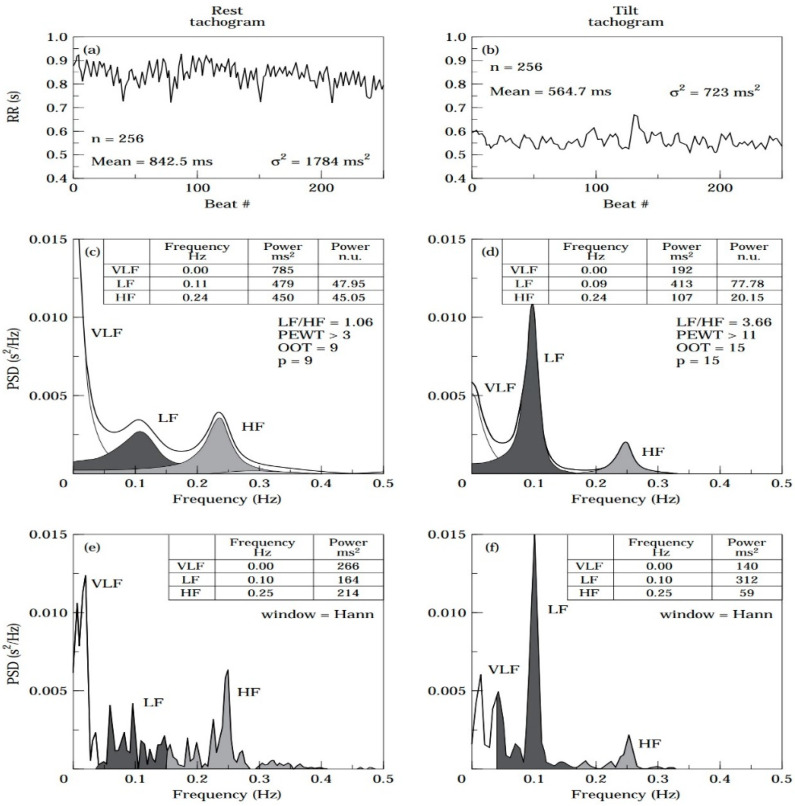
A power spectral density analysis on a normal individual at supine rest and after a head-up tilt. Source: [7].

**Figure 4 ijerph-20-07146-f004:**
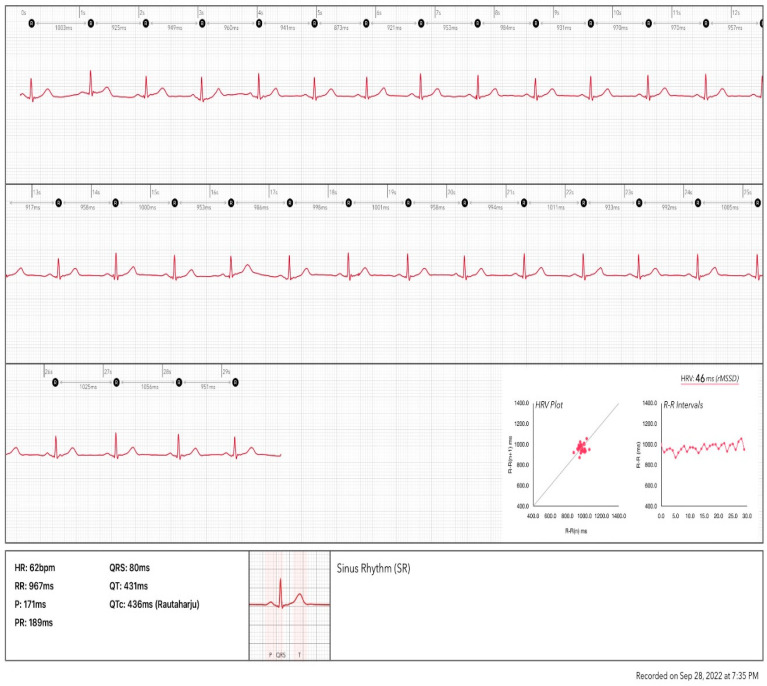
HRV analysis based on an Apple watch (series 7)-acquired ECG recording. The analysis included beat by beat NN intervals. The HRV is calculated as 47 ms using RMSDD. (This analysis was performed by the authors using the ECG HRV app on the iPhone 13).

**Figure 5 ijerph-20-07146-f005:**
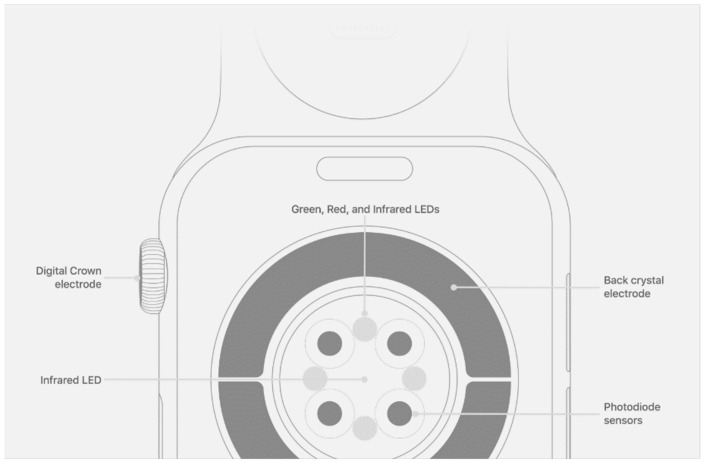
The PPG sensors on the Apple Watch that uses infrared sensors, photodiodes, and green LEDs to measure heart rate. The Apple Watch uses green LED lights paired with light-sensitive photodiodes to detect the amount of blood flowing through the wrist at any given moment. When the wearer’s heart beats, the blood flow in the wrist and the green light absorption are greater. Between beats, the absorption is less. By flashing its LED lights hundreds of times per second and calculating the green light absorption variation, the Apple Watch can calculate the heart rate. Source: [26].

**Figure 6 ijerph-20-07146-f006:**
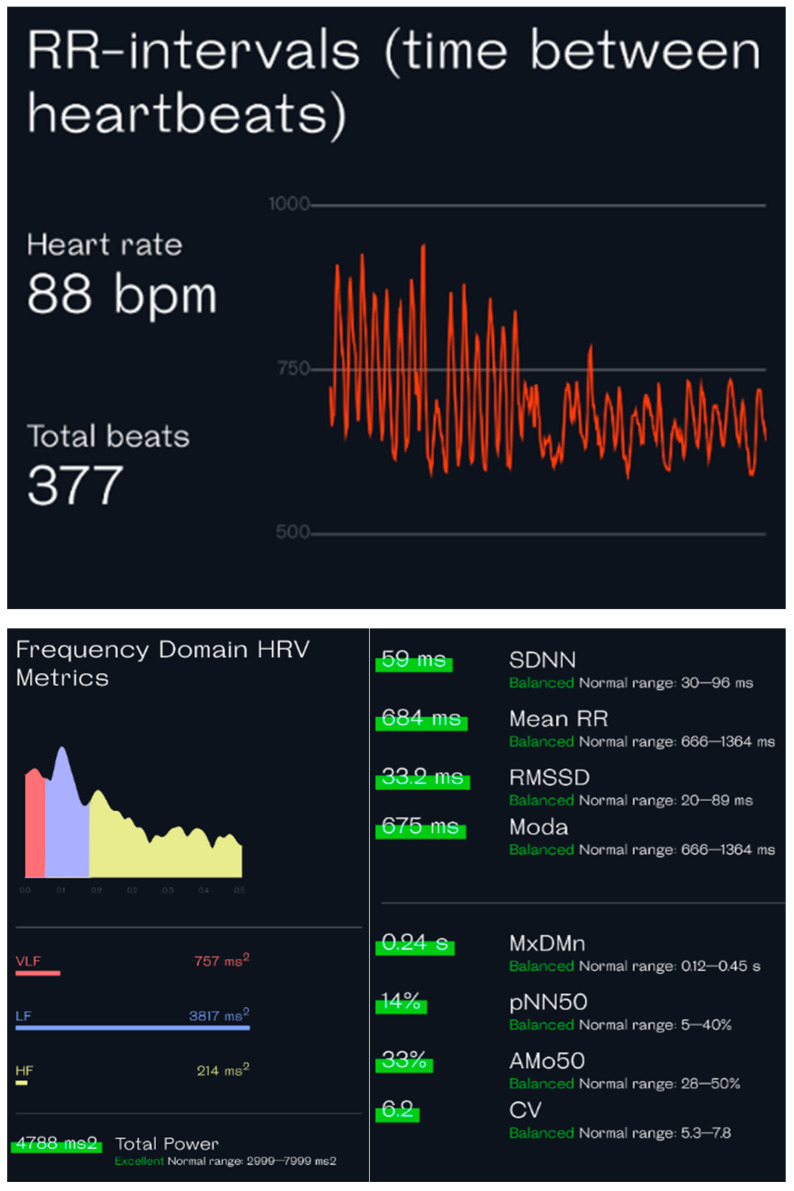
HRV analysis from the third-party app Welltory on iOS, which reads the raw data from a 5 min breath reading through an Apple Watch series 7. The (**upper panel**) shows the average heart rate and total heartbeats with a distribution of RR intervals. The (**lower panel**) shows the common time-domain measurements (**left side**) and the frequency-domain measurements (**right side**). (Analysis performed by the authors).

**Figure 8 ijerph-20-07146-f008:**
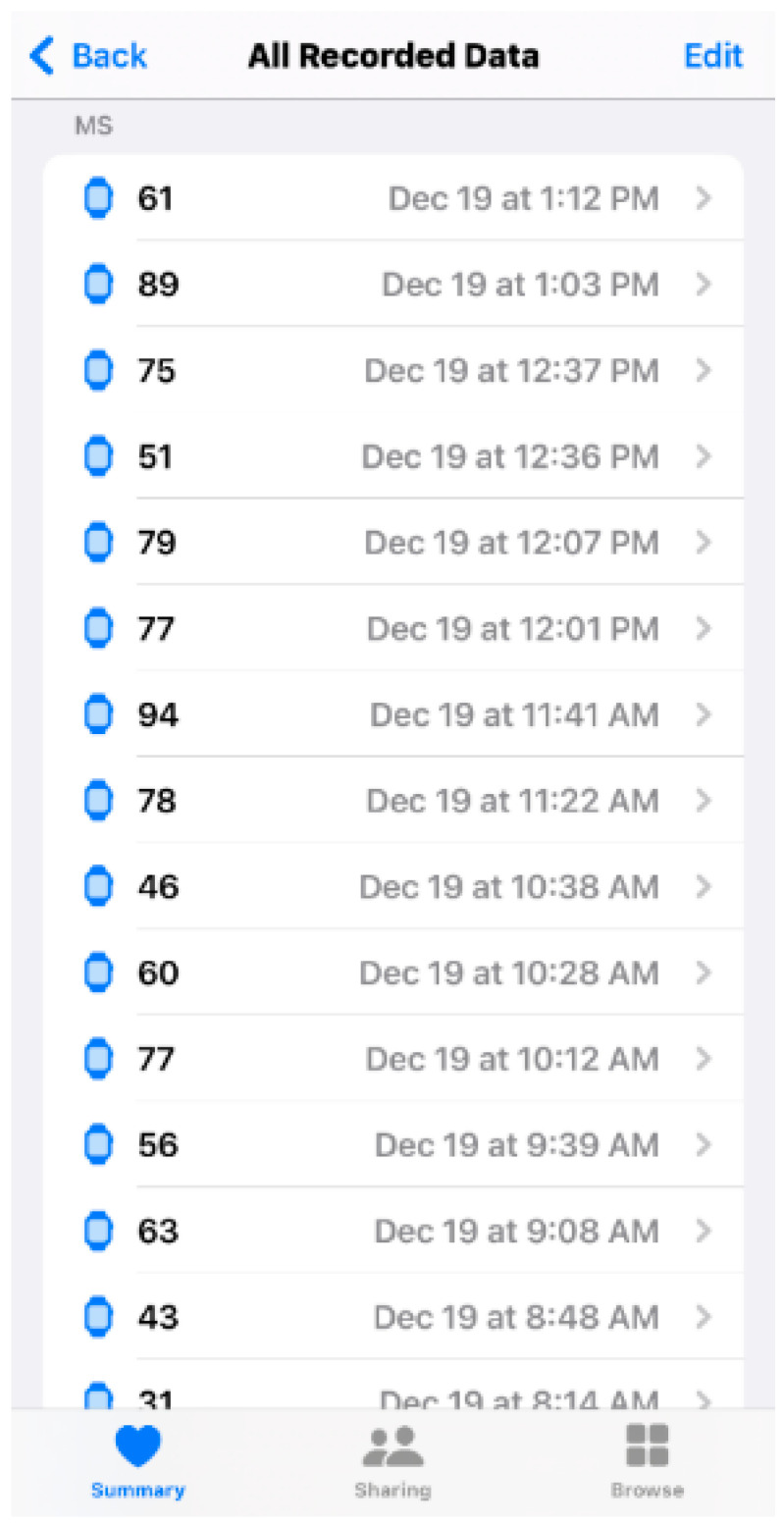
Heart rate variability data from Apple Watch automatic tracking in the Apple Health App (from authors’ data). Because the time points are randomly distributed, it is difficult to ascertain the wearer’s activity at the time of the recording.

**Figure 9 ijerph-20-07146-f009:**
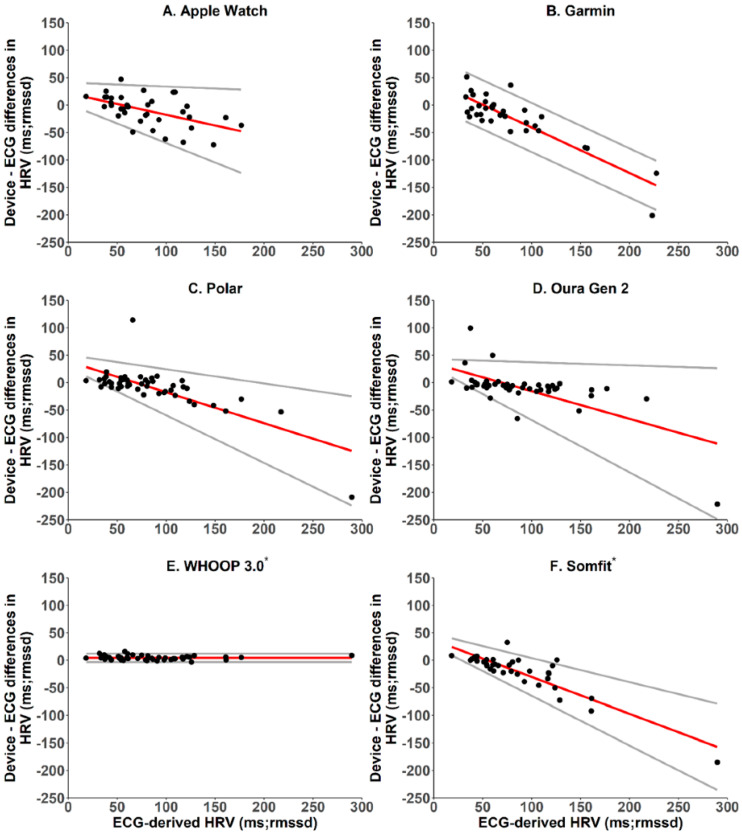
Bland-Altman plots of ECG-derived and PPG-derived heart rate variability measurements (from six brands). The measurements took place simultaneously, and different brands presented different reading variations. Source: [60]. * means manufacturers provided raw data.

**Figure 10 ijerph-20-07146-f010:**
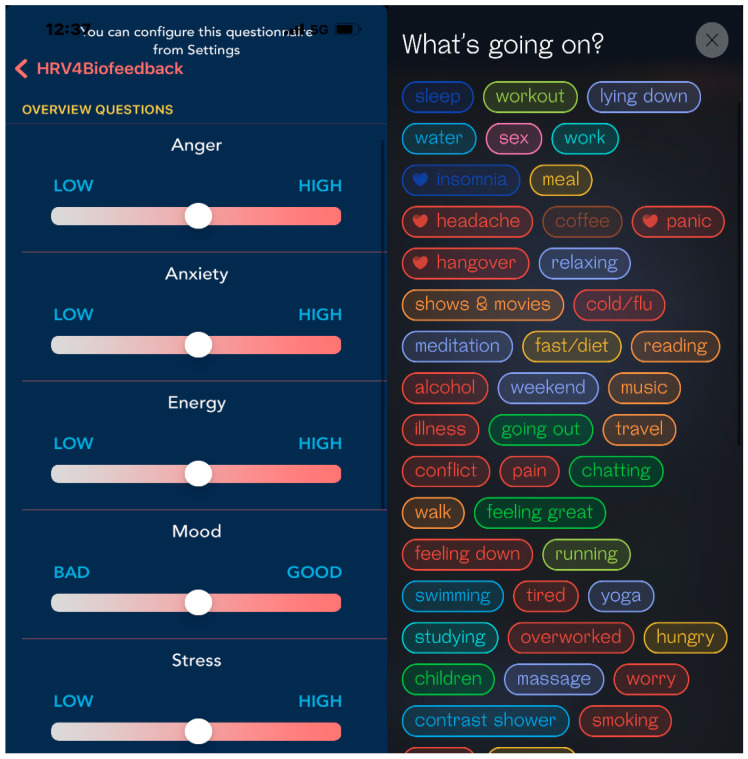
(**Left panel**): HRV4biofeed asks users to choose their current mental status before a manual session. (**Right panel**): after the manual session, the Welltory app asks users to choose the words that best reflect their mental status and activities (from the author’s iOS app).

**Table 2 ijerph-20-07146-t002:** Major wearable device brands that offer HRV monitoring.

Device	Recording Method	Recording Period	Recording Length	Analysis
Apple Watch	PPG	Automatic when the wearer works out	1 min	SDNN over the entire recording
		Automatic when the wearer starts the Breathe app	Determined by user, up to 5 min	SDNN over the entire recording
		Automatic every 10–15 min when the wearer enables the AF history	1 min	SDNN over the entire recording
Fitbit Watch/Band	PPG	Automatic when the wearer sleeps	Whole sleep	RMSDD over the entire recording
Garmin Watch	PPG	Automatic when the wearer sleeps	Whole sleep	RMSDD for every 5 min length
Oura Ring	PPG	Automatic when the wearer sleeps	Whole sleep	RMSDD for every 5 min length
		Automatic when the wearer starts unguided sessions	Determined by user, up to 180 min	RMSDD over the entire recording
Whoop Band	PPG	Automatic when the wearer sleeps	Whole sleep	Average of RR intervals over the entire recording
		Automatic when the wearer switches the mode to “broadcasting” and connects to third party HRV apps	Determined by user	Beat-to-beat intervals
AIO Smart Sleeve	ECG	Manual start by the wearer	Determined by user	Beat-to-beat intervals
Firstbeat Sport Sensor	ECG	Manual start by the wearer	Determined by user	Beat-to-beat intervals
Polar H10 Strip	ECG	Manual start by the wearer	Determined by user	Beat-to-beat intervals
Zephyr Bioharness	ECG	Manual start by the wearer	Determined by user	Beat-to-beat intervals

## Data Availability

No data were generated in this study.

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
