# Peer review of "Heart Rate Variability Measurement through a Smart Wearable Device: Another Breakthrough for Personal Health Monitoring?"

_ijerph, 2023, doi:10.3390/ijerph20247146_

Round 1

Reviewer 1 Report

Comments and Suggestions for Authors

The submitted manuscript reviews the current state of knowledge in heart rate variability (HRV) analysis based on wearable devices. The topic is significant in the context of the development of new wearable devices that record and analyze vital signs, including heart rate variability, however, the authors should consider using a more formal style and address the following comments:

Major:

-What was your search strategy, inclusion and exclusion criteria for including the studies and devices in the review?

-Why did your review concentrate only on commercially available wearable devices?

-If you consider experimental devices too, I would suggest including the devices based on mechanocardiography (ballistocardiography, seismocardiography, gyrocardiography).

Minor:

-I would rename the title of section 2 Introduction to Heart Rate Variability as "Heart Rate Variability"

-The formatting of the main part of the text should be uniform (one font size, one font typeface etc.)

-The part of the caption of the Figure 1: "Cited from Politano L, Palladino A, Nigro 87
G, et al. Usefulness of heart rate variability as a predictor of sudden cardiac death in muscular dystrophies. Acta Myol. 2008;27(3):114-122 [12]" could be rewritten as "Source: [12]". The rest of the citations of figures should follow the same correction type, e.g.:

The part of the caption of Figure 8: "Cited from Miller DJ, Sargent C, Roach GD. A Validation of 548
Six Wearable Devices for Estimating Sleep, Heart Rate and Heart Rate Variability in Healthy 549
Adults. Sensors. 2022; 22(16):6317" could be rewritten as "Source: [37]"

-Figures are blurred and significantly affected by lossy compression.

Comments on the Quality of English Language

The English language quality is satisfactory but the style of the text could be more formal.

Author Response

Major:

-What was your search strategy, inclusion and exclusion criteria for including the studies and devices in the review?

Response: We thank the Reviewer for the excellent question. We have now added a short Methods type section to more clearly describe our search strategy and inclusion and exclusion criteria (see Section 3; pages 2-3).

-Why did your review concentrate only on commercially available wearable devices?

Response: We concentrated on commercially available devices primarily because of the substantial, large user base, which facilitates large-scale data collection and analysis. Furthermore, this paper aims to engage the general population to show the advantages and usefulness of using these wearable devices to self-monitor health status.

-If you consider experimental devices too, I would suggest including the devices based on mechanocardiography (ballistocardiography, seismocardiography, gyrocardiography).

Response: We agree that those experimental devices are interesting and exciting, but we did not include them because the focus of our paper is on devices that are commercially available as indicated in our newly included inclusion Methods section for the review.

Minor:

-I would rename the title of section 2 Introduction to Heart Rate Variability as "Heart Rate Variability"

Response: We appreciate the Reviewer’s suggestion and have complied.

-The formatting of the main part of the text should be uniform (one font size, one font typeface etc.)

Response: We apologize for any oversight and have now ensured uniform formatting.

-The part of the caption of the Figure 1: "Cited from Politano L, Palladino A, Nigro 87
G, et al. Usefulness of heart rate variability as a predictor of sudden cardiac death in muscular dystrophies. Acta Myol. 2008;27(3):114-122 [12]" could be rewritten as "Source: [12]". The rest of the citations of figures should follow the same correction type, e.g.:

Response: We have now complied with the suggested formatting.

The part of the caption of Figure 8: "Cited from Miller DJ, Sargent C, Roach GD. A Validation of 548
Six Wearable Devices for Estimating Sleep, Heart Rate and Heart Rate Variability in Healthy 549
Adults. Sensors. 2022; 22(16):6317" could be rewritten as "Source: [37]"

Response: We have now addressed this issue in the caption for Figure 8.

-Figures are blurred and significantly affected by lossy compression.

Response: We apologize for any issues with the figures and have now added images with better resolution to the manuscript.

Reviewer 2 Report

Comments and Suggestions for Authors

The paper presents a valuable review of wearable heart rate variability (HRV) monitoring systems. Major consumer device brands and applications are listed and compared. Also, verification and different application aspects of wearable HRV devices are covered. Finally, advantages, challenges and limitations of wearable HRV devices are considered.

Minor comments:

The resolution of most of the figures has to be improved, most of them are blurred.

Too large font of some letters exists in line 81, 89, 172.

Please provide hyperlinks for all mentions applications in the paper, this will be useful for readers.

Author Response

Minor comments:

The resolution of most of the figures has to be improved, most of them are blurred.

Response: We apologize for the quality of the images, and we have now added figures with improved resolution.

Too large font of some letters exists in line 81, 89, 172.

Response: We apologize for any oversight in formatting and have addressed the issue.

Please provide hyperlinks for all mentions applications in the paper, this will be useful for readers.

Response: We appreciate the Reviewer’s helpful suggestion. We have now added the hyperlinks where appropriate.

Reviewer 3 Report

Comments and Suggestions for Authors

The authors presented interesting review on heart rate variability measurement through a wearable device for health monitoring. I have some comments:

1) The HRV estimated from the photosensor is more properly called pulse rate variability (PRV). A great deal of work has been devoted to the issue of comparing PRV and HRV and this aspect needs much more discussion in this review. In ideal conditions indeed many authors demonstrate quite high similarity (but not identity) of estimates obtained from HRV and PRV. However, when vascular biomechanical properties change (in pathologies or due to external factors), it causes a significant increase in the variability of the delay between beat-to-beat events in HRV and PRV, chaotically altering estimates, including spectral and other. Obviously this does not apply to gadgets based on HRV recording via unipolar ECG.

2) Despite the limitations described above, assessment of peripheral blood flow by a portable photosensor can provide other useful information (see DOIs: 10.1016/j.bpj.2021.05.020, 10.1038/s41746-019-0136-7, 10.3390/jcm9041203, etc.). This aspect is also worth mentioning briefly in the review.

3) The authors address aspects of ultra-short HRV analysis in too short. The problem of comparability of methods of HRV analysis based on ultra-short and standard ECG recordings is worth discussing in more detail. The applicability of usHRV instead of standard HRV in mass practice is actual topic.

4) Clearly state your position regarding the HRV analysis in patients with current AF.

5) It is worthwhile to pay more attention to promising measures for assessing chaotic and nonlinear dynamics in HRV (see DOIs: 10.1063/1.5134833, 10.2174/1874192400903010110, etc.).

Author Response

1) The HRV estimated from the photosensor is more properly called pulse rate variability (PRV). A great deal of work has been devoted to the issue of comparing PRV and HRV and this aspect needs much more discussion in this review. In ideal conditions indeed many authors demonstrate quite high similarity (but not identity) of estimates obtained from HRV and PRV. However, when vascular biomechanical properties change (in pathologies or due to external factors), it causes a significant increase in the variability of the delay between beat-to-beat events in HRV and PRV, chaotically altering estimates, including spectral and other. Obviously this does not apply to gadgets based on HRV recording via unipolar ECG.

Response: We appreciate the Reviewer’s suggestion and have now added two paragraphs on the topic of PRV in the manuscript as well as several supporting references (page 7, paragraphs 3 and 4).

2) Despite the limitations described above, assessment of peripheral blood flow by a portable photosensor can provide other useful information (see DOIs: 10.1016/j.bpj.2021.05.020, 10.1038/s41746-019-0136-7, 10.3390/jcm9041203, etc.). This aspect is also worth mentioning briefly in the review.

Response: We have now addressed this issue in the revised manuscript (lines 289-297).

3) The authors address aspects of ultra-short HRV analysis in too short. The problem of comparability of methods of HRV analysis based on ultra-short and standard ECG recordings is worth discussing in more detail. The applicability of usHRV instead of standard HRV in mass practice is actual topic.

Response: The use of usHRV was mentioned in discussing the Apple watch. Although we agree with the Reviewer that this topic is indeed worth exploring further, we believe that there is a lack of more substantial data in the current research involving the use of smart wearable devices and usHRV.

4) Clearly state your position regarding the HRV analysis in patients with current AF.

Response: We thank the Reviewer for this comment, and we have added a paragraph to the manuscript in regard to HRV analysis in patients with AF as well as supporting references (page 8, paragraph 2).

5) It is worthwhile to pay more attention to promising measures for assessing chaotic and nonlinear dynamics in HRV (see DOIs: 10.1063/1.5134833, 10.2174/1874192400903010110, etc.).

Response: As per the Reviewer’s suggestion, we have now added two paragraphs and the supporting references describing chaotic and nonlinear dynamics in HRV (bottom of page 4).

Reviewer 4 Report

Comments and Suggestions for Authors

I believe that the review addresses an important and current topic. However, there are some details that should be reconsidered:

1- There are many statements in the text that are not cited

2-The quality of the images must be reviewed

3- the commercial information that allowed you to choose these devices over others should be included with your bibliographic reference, perhaps in table 2.

4- The devices and when their principle of operation is explained must be cited with their web pages or with their invention patents.

5- Information on the use of Artificial Intelligence in the HRV systems of the devices evaluated should be included in Table 2.

6- I recommend putting the number of publications on HRV to date according to some database (Example: Pubmed), perhaps in the form of a graph.

Authors should review the notes of the resubmitted document for more details.

Author Response

1- There are many statements in the text that are not cited

Response: We apologize for any oversights in providing references in the manuscript. As per the comment, we have now carefully reviewed the revised manuscript and added references throughout the paper to support our text.

2-The quality of the images must be reviewed

Response: We apologize for the poor quality of the images and have now added figures with improved resolution.

3- the commercial information that allowed you to choose these devices over others should be included with your bibliographic reference, perhaps in table 2.

Response: We chose the devices to include in the review based on their availability to the general public for purchase and use; this approach aligns with the focus of the review, which is to show the advantages and usefulness of using these wearable devices to self-monitor health status.

4- The devices and when their principle of operation is explained must be cited with their web pages or with their invention patents.

Response: We appreciate the Reviewer’s helpful suggestion. Accordingly, we have added hyperlinks to Table 2.

5- Information on the use of Artificial Intelligence in the HRV systems of the devices evaluated should be included in Table 2.

Response: We have described how HRVs are calculated in each device based on the methods we mentioned in Table 1. The calculation methods are fixed for each device, and no AI is needed. AI may play a role in deciding when to start recording. Based on the current technology, most devices need the wearer to decide when to start recording or record at the default time, such as during sleep, as we listed in Table 2. Only the Apple Watch has an automatic recording function that may have some kind of "AI" to decide the recording time point, but there is no detail on how it chooses the time point, as we discussed in Figure 8.

6- I recommend putting the number of publications on HRV to date according to some database (Example: Pubmed), perhaps in the form of a graph.

Response: As per the Reviewer’s suggestion, we have now added a new figure (Figure 2) to show the HRV publications to date from the PubMed database.

Round 2

Reviewer 1 Report

Comments and Suggestions for Authors

The authors have addressed my comments. However, I have a few suggestions that may improve the quality of the paper before it is accepted for publication:

1. Proofreading the whole text to check minor mistakes, e.g. punctuation, wording etc.

2. Table 2

- I would rename the column "Brand" into "Device"

-You do not have to repeat the recording method for each row if it is the same for each recording period unless it changes (or the device changes)

3. Figure 7: "Cited from: [full citation]" is unnecessary in the caption if you use a numbered reference. You can use "Source: [52]".

Comments on the Quality of English Language

The quality of English is acceptable, however, minor proofreading would be beneficial for the overall quality of the submitted manuscript.

Author Response

The authors have addressed my comments. However, I have a few suggestions that may improve the quality of the paper before it is accepted for publication:

  1. Proofreading the whole text to check minor mistakes, e.g. punctuation, wording etc.
  • The text has been professionally edited. The document has been reviewed, and the grammar and syntax are correct.

  1. Table 2

I would rename the column "Brand" into "Device"

  • We appreciate the Reviewer’s suggestion, and we have edited the table accordingly.

You do not have to repeat the recording method for each row if it is the same for each recording period unless it changes (or the device changes)

  • We have edited the table as per the comment.

  1. Figure 7: "Cited from: [full citation]" is unnecessary in the caption if you use a numbered reference. You can use "Source: [52]".
  • We have complied with the Reviewer’s suggestion.

Reviewer 3 Report

Comments and Suggestions for Authors

I approve of the current version of the article.

Author Response

NONE